# A *b* map implying the first eastern rupture of the Nankai Trough earthquakes

K.Z. Nanjo [1] & A. Yoshida[2]

The Nankai Trough megathrust earthquakes inflicted catastrophic damage on Japanese society and more widely. Most research is aimed at identifying strongly coupled regions that are considered as a major source of future disastrous earthquakes. Here we present a *b*-value map for the entire Nankai Trough zone. The *b* value, which represents the rate of occurrence of small earthquakes relative to larger ones, is inversely dependent on differential stresses, and has been used to detect highly stressed areas on fault planes in various tectonic situations. A remarkable finding is that the *b* value is inversely correlated with the slip-deficit rate (SDR). Moreover, the *b* value for the areas of high SDR in the eastern part is lower than that in the western part, indicating that differential stress on asperities in the eastern part is higher than that in the western part. This may explain the history of the Nankai Trough earthquakes, in which the eastern part tends to rupture first.

[1] Global Center for Asian and Regional Research, University of Shizuoka, 3-6-1, Takajo, Aoi-Ku, Shizuoka 420-0839, Japan. [2] Center for Integrated Research and Education of Natural Hazards, Shizuoka University, 836, Oya, Suruga-Ku, Shizuoka 422-8529, Japan. Correspondence and requests for materials should be addressed to K.Z.N. (email: nanjo@u-shizuoka-ken.ac.jp) or to A.Y. (email: akio.yoshi@nifty.com)

The occurrence of the next Nankai Trough earthquake will initially be observed by high-sensitivity geophysical monitoring networks, such as Hi-net[1], GEONET[2], DONET[3], and GPS-A[4, 5]. Various precursory phenomena are expected to appear before this occurrence. Possible candidates[6–10] include, but are not limited to, reported precursors to the 2011 Tohoku-oki megathrust earthquake of magnitude (M) 9, such as weakening of interplate coupling, seismic quiescence and activation, tidal triggering, and changes in the relative occurrence of large and small events. In order to detect these phenomena, steady and intense monitoring of seismicity and crustal movement is indispensable. A number of studies have been conducted aimed at identifying strongly coupled regions that should be a major source of released seismic energy in future disastrous earthquakes because megathrust earthquakes are driven by interplate coupling[6, 11–18]. Focal areas of historical earthquakes are thought to be located around high slip-deficit-rate (SDR) regions on the plate interface[19, 20], where differential stress is high[21, 22]. Therefore, to search high SDR and highly stressed areas is most important. Yokota et al.[23] elucidated the distribution of SDRs in the Nankai Trough subduction zone by analyzing seafloor as well as land geodetic observation data obtained from GPS-A and GEONET. A b-value analysis exhibits spatial distribution of differential stress. The cumulative number of earthquakes with a magnitude larger than or equal to M is well approximated by the Gutenberg–Richter (GR) law[24] as follows: $\log_{10} N = a - bM$, where a and b are constants (i.e., a high b value indicates a larger proportion of small earthquakes, and vice versa). Spatial and temporal changes in b are known to reflect structural heterogeneity, strength and temperature of the seismogenic zone[22, 25–27]. In the laboratory, as well as in the Earth's crust, the b value is also known to be inversely dependent on differential stresses[28, 29].

In this context, the b value is considered to be a proxy of stress which could help designate asperities or highly stressed patches in the plate interface where future ruptures are likely to occur[21, 22, 30].

Similar to the studies by Nanjo et al.[21] and Tormann et al.[22], we apply the b-value analysis to the plate boundary region between the Philippine Sea and Amur plates, where megathrust earthquakes are expected to occur in the near future. Although the b value is spatially heterogeneous, we find that it is low in the focal areas of the M8-class 1944 Tonankai and 1946 Nankai earthquakes. Combining SDRs with the b values, we propose that the most stressed region is the eastern part of the 1944/1946 earthquakes. Furthermore, we argue that this is the most likely nucleation region for the next large event, which would be consistent with the reported rupture history.

## Results

**A b-value map of the Nankai Subduction.** Our study creates a b map for the entire Nankai Trough, as shown in Fig. 1. For details on the data processing of earthquakes and the mapping procedure, see Methods and Supplementary Figs. 1–6, which provide comprehensive information on the methods and additionally performed sensitivity checks that confirm the robustness of the presented findings. A first glance of the map indicates a band of low b values (b < 0.7 in red) from northern Shikoku to the northern Ise Bay where the depth of the subducting slab is ~30 km or lower. Location of this band is on the deeper side of the focal regions of the 1944 Tonankai and 1946 Nankai earthquakes, which represent the last series of the Nankai Trough earthquake. Interestingly, the low b-value zone overlaps with the zone of low-frequency earthquakes[31, 32].

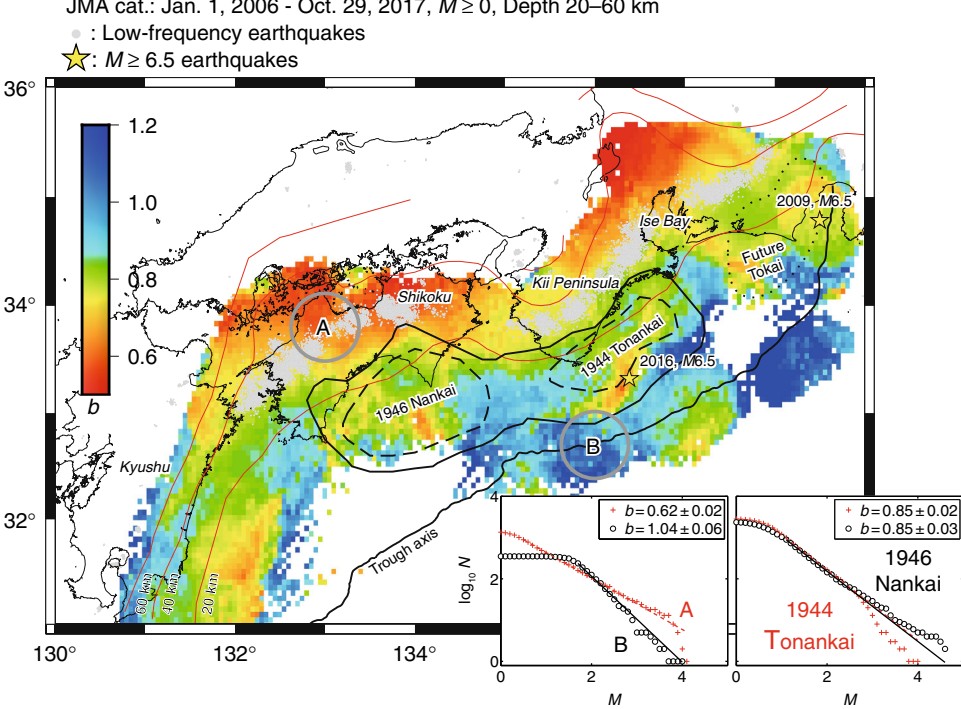

**Fig. 1** Map of b values. Earthquake data used to create this b map are given at the top left of this figure (also see Data availability). The software package ZMAP[47] was used to create a dense spatial grid (0.05×0.05 degrees), to sample earthquakes that fell in overlapping volumes of circular shape with a radius of r = 35 km, and to map b values calculated by the EMR[46] technique. Dotted line delineates a region supposed to be sources of the anticipated Tokai earthquake[14]. Solid and dashed lines denote the regions of estimated large slips (>2 and 4 m, respectively) resulting from the 1946 Nankai and 1944 Tonankai earthquakes[12]. Grey dots show low-frequency earthquakes. Red curves mark depth contour lines[40-43] (also see Data availability). Inset: frequency–magnitude distribution for annotated regions with r = 35 km indicated by A and B and for the slip regions (>4 m)

Low-frequency earthquakes are considered to occur at the plate interface, where effective differential stress is small and shear sliding occurs easily due to the presence of ample fluids that are supplied from the subducting slab[33, 34]. On the other hand, within the slab, from which fluids are extracted, "ordinary" earthquakes may occur in a drained condition, where effective differential stress should be high due to low pore-pressure, which is supposed to be the reason why the low $b$-value band overlaps with the zone of low-frequency earthquakes. This overlap on the deeper side of the coupled region seems to be a unique feature of the Nankai Trough, which is not seen on the down-dip side of the 2011 Tohoku-oki earthquake[22].

Patches with relatively high $b$ values ($b > 1$ in blue) in the zone along the Trough axis indicate that these areas are in a state of low differential stress. Due to low seismicity, the $b$ map is blank in the area far south off Shikoku, where it is believed that no slip occurred during the 1946 Nankai earthquake[12]. High $b$ values near the Trough axis show a common feature of both subduction zones of the Nankai Trough and Tohoku-oki[22].

In the intermediate area where the depth of the upper surface of the subducting slab lies between 10 and 30 km, the color of the $b$ map ranges from orange and yellow to green and cyan. Within this depth range, a relatively high $b$-value area ($b \sim 0.9$) is recognized south of the Kii Channel between large-slip regions ($>4$ m) of the 1944 Tonankai and 1946 Nankai earthquakes, indicating that the source regions of the two megathrust earthquakes are substantially separated. Variation of the $b$ value along the line that passes through estimated large-slip areas of the two earthquakes shows that the $b$ value at the parts between these areas is significantly high with a degree that surpasses the level of uncertainty (Supplementary Fig. 6). The high $b$-value region corresponds to the low-SDR region, Region C marked in Fig. 2 of Yokota et al.[23]. Both ruptures of the 1944 Tonankai and the 1946 Nankai earthquakes started from the high $b$-value region, the rupture of the former earthquake proceeded towards the east while the rupture of the latter earthquake extended to the west, to the south of Shikoku.

Existence of the two notable low $b$-value patches as well as the high $b$-value area between them is unique in the Nankai Trough subduction zone, suggesting that the plate interface is segmented in such a way that two megathrust earthquakes might occur separately, thereby breaking each patch. This spatial pattern of the $b$ value is in contrast with that in the Tohoku-oki subduction zone[22], where the Pacific Plate subducts beneath the North-American plate.

**Relationship between $b$ value and SDR.** Interestingly, as is shown in Fig. 2, we found a clear inverse relationship between $b$ values and SDRs (for the data processing of SDRs, see Methods). When calculating the $b$ value, we sampled events that fell within a cylindrical volume with radius $r$, centered at each node with a SDR value. We used radii $r = 6$ and $8$ km. For details of the procedure used to make the graph of $b$ values versus SDRs, see Methods and Supplementary Figs. 7-9. Figure 2a shows the result for the entire coupling zone along the Nankai Trough, where an inverse relationship between the $b$ value and SDR can be seen, indicating that differential stress is high in large SDR areas. This feature is more pronounced for a small $r$. This is reasonable because sampling from a larger volume is considered to smooth out differences in $b$ for different SDRs. We examined if the observed inverse relationship between $b$ values and SDRs can occur contingently using randomized catalogs and found that it is very unlikely to occur (see Methods and Supplementary Fig. 9).

It is known that the eastern and western parts of the historical series of Nankai Trough earthquakes often ruptured

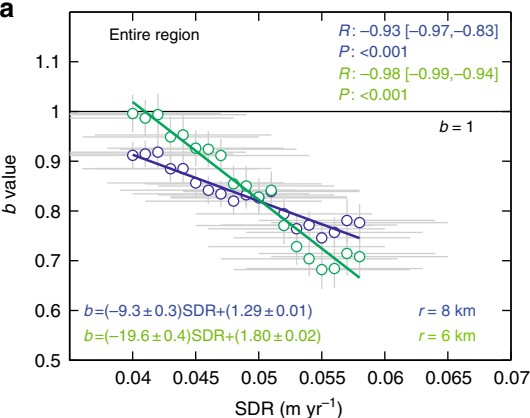

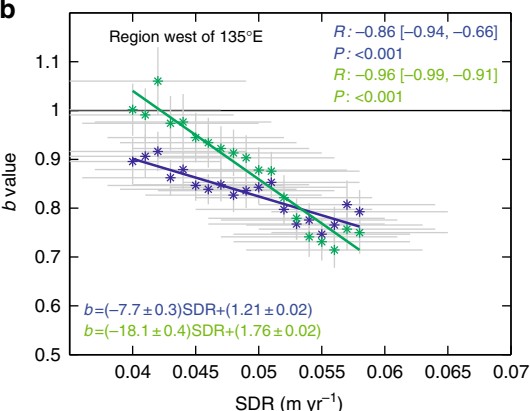

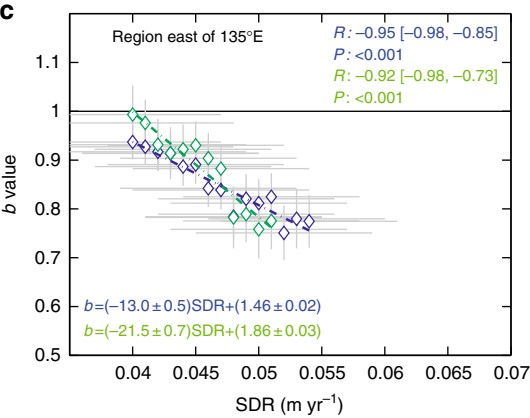

**Fig. 2** Plot of $b$ values as a function of SDR. Horizontal and vertical bars in gray indicate sliding windows (width of 0.014 m year⁻¹) and bootstrapping errors, respectively. The error bars represent one standard deviation of a bootstrap sampling distribution. We used radii: $r = 6$ and $8$ km in green and blue, respectively. The least-square regression lines are provided in the corresponding color. $R$: correlation coefficient with the lower and upper bounds in brackets for a 95% confidence interval for the coefficient, $P$: P-value of a significant test. We divided **a** the entire region into **b** the region west of 135°E, and **c** the region east of 135°E

separately[6, 11-14]. In order to see if any difference could be observed between the eastern and western parts in the inverse relationship of $b$ and SDR, we investigated the relationship by dividing the entire region into two sub-regions on the 135°E longitude, which separates focal regions of the 1944 Tonankai and the 1946 Nankai earthquakes. From Fig. 2b, c, where the $b$ value is plotted against the SDR for each of the sub-regions, we observed

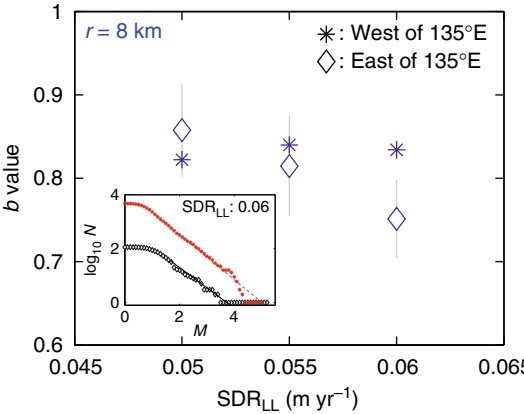

**Fig. 3** $b$ values for several values of $SDR_{LL}$. Vertical bars in gray indicate bootstrapping errors displaying one standard deviation of bootstrap uncertainty. We used samples of a radius of $r = 8$ km. Asterisks mark the result for the region west of 135°E, and diamonds for the region east of 135° E. Inset: frequency–magnitude distribution for $SDR_{LL} = 0.06$ m year$^{-1}$

that the inverse relation between $b$ and SDR was retained for both regions (see also Supplementary Fig. 7).

It is remarkable, however, that at large SDRs that lie below the plate convergent rate (SDR ≤ 0.065 m year$^{-1}$), the $b$ values in the eastern part are lower than those in the western part. This difference in $b$ is highlighted in Fig. 3, where $b$ values are plotted for SDRs above a lower limit $SDR_{LL}$, which is 0.05, 0.055, and 0.06 m year$^{-1}$. In obtaining the results, we collected $M \geq 0$ earthquakes, using a sampling radius $r = 8$ km, centered at each node with $SDR \geq SDR_{LL}$, avoiding a double-count of the same events. The difference was more pronounced for a larger $SDR_{LL}$. When a small radius $r = 6$ km is adopted, the calculation could not be carried out, because of insufficient data (Supplementary Fig. 10). On the other hand, for a larger sampling radius $r = 10$ km, the difference between the eastern and western parts was not significant as the $b$ value is considered to be smoothed out by contamination of the surrounding points with smaller SDRs.

## Discussion

Our finding, for large SDR areas, that the $b$ value in the eastern part is apparently lower than that in the western part (Fig. 3) is somewhat intriguing because it indicates that differential stress on asperities in the eastern part is higher than that in the western part, even if SDRs on the asperities are the same. This finding can be understood if we note, from Fig. 2 in Yokota et al.[23], that the dimension of the areas of large SDRs in the eastern part is smaller than that in the western part (also see Supplementary Fig. 8), which indicates that the rate of rising stress on the asperities in the eastern part will be larger than in the western part. The higher rate of rising differential stress on the asperities in the eastern part is probably the reason for the observed tendency in which eastern part ruptured first in the historical series of megathrust earthquakes[6, 11–14].

The SDR-model[23], which was based on geodesy data, elucidated spatial characteristics of the coupling condition in the Nankai Trough, and identified strongly coupled (high SDRs) areas and weakly coupled (low SDRs) areas. On the other hand, we showed, by creating a $b$ map in Fig. 1, where the differential stress is high (low $b$ value) and where it is low (high $b$ value). Although these two methods are based on different data and ideas, it was found that the $b$ value is correlated with SDR: the larger the SDR, the lower the $b$ value (Fig. 2).

Wiemer et al.[35] demonstrated that transients in subsidence rate in the Tokai region, specifically the eastern Nankai Trough zone,

were correlated with a change in the occurrence rate and size distribution of earthquakes, i.e., the $b$ value. They surmised that an increase in the locking stress, namely a "slow stick event", was consistent with all observations. Ghosh et al.[36] observed an inverse relation between the $b$ value and locking for the subducting Cocos Plate beneath Costa Rica. An observation from the Parkfield segment of the San Andreas Fault showed significantly different $b$ values during a period of aseismic transient events, corresponding to unlocking a part of that segment[27]. This phenomenon was studied in more detail by Tormann et al.[37] who demonstrated a correlation between the time-series of $b$ values and variation in creep rate. When all of these observations are taken into account, it is reasonable to consider that the low $b$ value in areas of large SDR indicates that differential stress is large on asperities.

The eastern and western highly stressed asperities (low $b$ values) at depths of 10 to 30 km of the subducting plate almost correspond to the large-slip regions of the 1944 Tonankai and the 1946 Nankai earthquakes, respectively. Although the analyzed period in this study is 1/10 or shorter than the seismic cycles, we believe that the separation of highly stressed asperities is an intrinsic feature of the subduction zone along the Nankai Trough, and may explain why ruptures are often segmented to the eastern and western parts in the historical series of megathrust earthquakes[6, 11–14]. Our finding that differential stress at large SDRs in the eastern part is higher than that in the western part (Fig. 3) may suggest that the eastern part will rupture earlier than the western part also in the next series of earthquakes, although we cannot say how long the time-lag will be.

Nanjo et al.[21] showed that the $b$ value in the large-slip area of the 2011 $M$9 Tohoku-oki earthquake had decreased notably preceding its occurrence. The dimension of the asperity was ~1×1°, in which the $b$ value dropped to <0.5 just before the earthquake. Currently, such areas, in which the $b$ value has been continuously decreasing, are not recognized in the coupling zone along the Nankai Trough (in the depth range between 10 and 30 km). Although the quantitative predictive power of a $b$ map is sometimes questioned, as in the case of the 2003 $M$8 Tokachi-oki earthquake[22], it is certain that the $b$ value became small in the case of the 2011 $M$9 Tohoku-oki earthquake[21]. There is no doubt that a transient change in $b$ provides significant information about the evolution of stress in the plate coupling region and that a decrease in the $b$ value is a prospective precursor to the next interplate earthquake. Therefore, we consider that monitoring the $b$ value as well as SDR is indispensable to evaluate whether the next series of the Nankai Trough earthquakes is approaching or not.

## Methods

**Data processing of earthquakes**. The Japan Meteorological Agency (JMA) catalog includes earthquakes since 1923 (also see Data availability). To complement its own network, JMA started in 1997 real-time processing of waveform data from many other networks operated by Japanese universities and institutions. An improvement of the network was almost completed in the southwestern part of Japan in 2000 and in the northeastern part in 2002[1, 38]. Homogeneity of recording in the Nankai Trough subduction zone was ensured since 2000. Currently, JMA, in collaboration with Ministry of Education, Culture, Sports, Science and Technology, processes waveform data from networks operated by Hokkaido University, Hirosaki University, Tohoku University, the University of Tokyo, Nagoya University, Kyoto University, Kochi University, Kyushu University, Kagoshima University, Yokohama City, Aomori Prefecture, Shizuoka Prefecture, Tokyo Metropolitan Government, JMA, Geospatial Information Authority of Japan, National Research Institute for Earth Science and Disaster Resilience, National Institute of Advanced Industrial Science and Technology, Japan Agency for Marine-Earth Science and Technology, Association for the Development of Earthquake Prediction, and Hot Springs Research Institute of Kanagawa Prefecture.

In 2004, $M$7.4 and $M$7.1 earthquakes occurred within the Philippine Sea plate located off the Kii Peninsula near the Trough axis, which was a very unusual location. These earthquakes were followed by active aftershocks, which do not

provide a steady-state behavior in the $b$ value. To create a $b$ map, we avoided temporal changes in the $b$ value and did not included seismicity around the time of the large earthquakes. To achieve this, a data set since 2006 was used in $b$-map analysis (Fig. 1). SDR analysis was performed by using data in that period (see the section "Data processing of SDRs").

In depicting spatial variation of the $b$ value in the coupling zone, we used both earthquakes in the Philippine Sea plate and earthquakes at the interface between the Amur and Philippine Sea plates. This is because seismicity in the Philippine Sea plate in the coupling zone is considered to reflect interplate partial locking between the subducting plate and the overriding plate[39]. To select earthquakes in the Philippine Sea plate and on the plate interface, we created a 100 km-wide cross-section from 31°N to 36.0°N for several longitudes, 132°E up until 138°E (Supplementary Fig. 1), and found that almost all earthquakes in the Amur plate are distributed at depths of 0–20 km. The seismicity that becomes deeper from south to north is considered to represent earthquakes occurring in the Philippine Sea plate and at the plate interface (also see Data availability for plate configuration data[40–43] used in this study). Determination of the location of earthquakes occurring offshore (outside of the network) is poor, and thus earthquake hypocenters far-offshore are determined deeper than 20 km. This feature of the catalog influenced our decision to use earthquakes deeper than 20 km in our analysis. In the study region, two earthquakes with $M6.5$ occurred since 2006. One earthquake, which was located at the margin of sources of a future Tokai earthquake, occurred in the Philippine Sea plate in 2009[44]. The other earthquake, which was ~50 km offshore from the Kii Peninsula, occurred on 1 April 2016 at the plate interface[45].

Low-frequency earthquakes[31, 32] are also listed in the JMA catalog (Fig. 1). At depths of ~30 km, a band of low-frequency earthquakes are overlaid above "ordinary" earthquakes (Supplementary Fig. 1). Low-frequency earthquakes are considered to occur at the plate interface[33, 34] when water is plentiful, and this water is squeezed out from the subducting slab. Low-frequency earthquakes were not included in our $b$-value analysis, but they were compared with $b$ values of "ordinary" earthquakes.

**Mapping procedure**. To estimate $b$ values homogeneously over space and time, we employed the EMR (Entire-Magnitude-Range) technique[46], which also simultaneously calculates the completeness magnitude $M_c$, above which all events are considered to be detected by a seismic network, and $a$ value of the GR law. The software package ZMAP[47] was used to facilitate computing and mapping $b$ value based on the EMR method, as described in the next paragraph. EMR applies the maximum-likelihood method[48] when computing the $b$ value to events with a magnitude above $M_c$. We always computed a $b$ value for the corresponding sample only if at least $N_{min} = 50$ events yielded a good fit to the GR law. The insets in Fig. 1 show a good fit of the GR law to observations in the present cases.

The code of the EMR method is freely available together with the seismicity analysis software package ZMAP[47] (http://www.seismo.ethz.ch/static/stat_2010_website/stat-website-pre2010/www.earthquake.ethz.ch/software/zmap.html), which is written in Mathworks' commercial software language Matlab® (http://www.mathworks.com). No knowledge of the Matlab language is needed since ZMAP is GUI-driven, although the ZMAP code is open. The ZMAP combines many standards seismological tools. Evaluating spatial variations in seismicity is one of the primary research objectives of ZMAP. By creating a dense spatial grid and sampling overlapping volumes of circular shape, ZMAP users can map $b$ values calculated by the EMR[46]. Through this study, we used a grid spacing of $0.05 × 0.05°$ and a sampling radius $r = 35$ km for mapping $b$ values, except for Supplementary Fig. 2 that shows $b$ maps created for different radii $r$ to identify the best representative among those maps.

The EMR technique[46] was initially proposed by Ogata and Katsura[49, 50], who combined the GR law with a detection rate function. Statistical modeling was performed separately for completely detected and incompletely detected parts of the frequency–magnitude distribution. The $b$ and $a$ values in the GR law are computed based on earthquakes above a certain magnitude ($M_{cc}$). For earthquakes whose magnitudes are smaller than $M_{cc}$, it has been hypothesized that the detection rate depends on their magnitudes in such a way that large earthquakes are almost entirely detected while smaller ones are detected at lower rates. Several studies[46, 49, 50] assumed that the detection rate was expressed by the cumulative function of the Normal distribution. Earthquakes with magnitudes greater than or equal to $M_{cc}$ are assumed to be detected with a detection rate of one. To evaluate the fitness of the model to data, the log-likelihood is computed by changing the value of $M_{cc}$. The best fitting model is that which maximizes the log-likelihood. Woessner and Wiemer[46] pointed out that the choice of the cumulative function of the Normal distribution, based on visual inspection and modeling of a variety of catalogs, as well as comparisons to other possible functions, is problematic because it is not based on physical reasoning. They suggested that such cases might exist when the choice of another function is more appropriate. However, synthetic tests endorsed that estimates of $M_c$ can be correct even if the cumulative function of the Normal distribution is replaced by the cumulative function of the Lognormal or Weibull distribution.

We searched the optimal sampling volume (Fig. 1) by mapping $b$ values with a wide range of radii $r$ and selected the largest radius that provided the most detailed resolution of the $b$ value heterogeneity (inhomogeneity). The observation of a nearly identical pattern of $b$ values when sampled with radii of 20 km to 35 km

suggests that using radii smaller than 35 km only reduces coverage (Supplementary Fig. 2). Sampling with radii greater than 35 km resulted in smoothed $b$ values. Sampling with a radius of $r ≥ 40$ obscured any $b$ value contrast. Thus, the appropriate radius of the volumes is about 35 km, because sampling with smaller radii reduces coverage while sampling with larger radii obscures anomalies and contrasts. In making $b$ maps through this study, we sampled earthquakes within a radius of $r = 35$ km (Fig. 1). The EMR technique[46] also calculates $M_c$ and $a$ simultaneously, thus the maps of $M_c$ and $a$, which were created when the $b$ map in Fig. 1 was obtained, are shown in Supplementary Fig. 3.

We took an alternative mapping approach by sampling nearest events to each node and computed a $b$ value for the corresponding node (Supplementary Fig. 4). We computed a $b$ value for the closest 200 earthquakes to the node when the sampling radius was $r ≤ 35$ km and only if at least $N_{min} = 50$ events yielded a good fit to the GR law. This feature is similar to that seen in Fig. 1, indicating that the general pattern remains stable with regards to sampling methods. A sparse map of $b$ values, taking $N_{min} = 100$ events, shows a similar pattern to the map.

A $b$-value analysis is critically dependent on a robust estimate of completeness of the processed earthquake data. In particular, underestimates in $M_c$ lead to systematic underestimates in $b$ values. We processed nearly $1.1 × 10^5$ earthquakes since 2006 with $M ≥ 0$ in the study and always paid attention to $M_c$ when assessing $b$ locally at each node (Supplementary Fig. 3c). As discussed in other studies using the JMA catalog[38], $M_c$ in offshore regions is expectedly higher than within the network constructed in onshore regions: $M_c$ gradually increases with distance from the coast (Supplementary Fig. 3c). There is a visual correlation between the gradients in $M_c$ and $b$: low-$M_c$/low-$b$ below the coast and increasing $M_c$ and $b$ going offshore towards the Trough axis. To eliminate any doubts whether this is biased, we conducted a quick additional robustness test. This was achieved by creating two $b$-value maps for increased values of every local $M_c$ by 0.2 and 0.5 magnitude units (Supplementary Fig. 5). We found that the pattern remains stable with reduced plotted area.

**Data processing of SDRs**. We used a data set of SDRs in the subduction zone along the Nankai Trough that was obtained from the Supplementary Information of the Yokota et al. study[23] (also see Data availability). Since 2000, the Hydrographic and Oceanographic Department of the Japan Coast Guard acquired seafloor geodetic data using a combined positioning system and acoustic ranging (GPS-A)[4, 5, 51]. Combining seafloor geodesy data since 2006 with onshore geodesy data, the Department created an offshore interplate SDR-model. The model is epoch-making in the sense that high SDR regions were clearly elucidated. Although many other geodetic studies have attempted to detect high SDR regions for the Nankai Trough, they have not been successful. This is because the previous geodetic observation network was biased towards land areas so that total geodetic information on the seafloor could not be obtained[15–17]. Although a small-scale seafloor geodetic observation has been carried out[18], it was limited to the region off the Kii Peninsula.

**Procedure used to make the graph of $b$ values versus SDRs**. We assumed a sliding window approach along SDR (Fig. 2 and Supplementary Fig. 7). Next, we stacked events for each window avoiding a double-count of the same events. We then removed from our analysis the effect of mixtures from different regions with heterogeneous completeness levels. To achieve this, we precut the earthquake catalog at $M2.2$ (completeness of the catalog for the entire Nankai Trough zone in Supplementary Fig. 3c is gained for earthquakes with $M2.2$ or larger). We then computed a $b$ value for the corresponding window.

We did not consider SDRs in regions north of 34.7°N. This is because, as discussed in the main text, the plate interface in these regions is not considered to be coupled and because Yokota and Watanabe[52] proposed a revised version of the SDR model, from which SDRs in regions north of 34.7°N, which were included in the original version[23], were excluded.

The $b$ value is given for the sampling radii $r = 6$ and 8 km as a function of SDR for the entire region of the Nankai Trough, the region west of 135°E, and the region east of 135°E in Fig. 2a, b, c, respectively, where the window width is 0.014 m year$^{-1}$ (for a narrower window width of 0.010 m year$^{-1}$ in Supplementary Fig. 7, we confirmed that the general pattern remains stable) and the step is 0.001 m year$^{-1}$ (the use of this value was determined by taking into account the significant figure of a data set of SDR in the Supplementary Information of the Yokota et al. study[23]). $b$ values were estimated based on seismicity in 1 January 2006–29 October 2017, adding data from the period from 1 January 2000 to 4 September 2004 (excluding off the Kii Peninsula seismicity) to enhance the steady-state behavior of $b$ values. We considered that the range of data points $0.04 ≤ SDR ≤ 0.058$ m year$^{-1}$ is meaningful. The upper limit of the SDR range is smaller than the convergence rate[23] (0.065 m year$^{-1}$) of the Philippine Sea plate under the Amur plate. The stress dependence of the $b$ value is different from that of SDR, in which coupling is expected above a certain value of stress but is absent below it. To separate this from the absence of coupling, we decided to use a higher value (0.04 m year$^{-1}$) than the detection limit of SDR[23] (0.03 m year$^{-1}$) as the lower limit of the range of SDRs (see Supplementary Fig. 8 for areas of $SDR ≥ 0.04$ m year$^{-1}$). An inverse relationship between the $b$ values and SDRs is characterized by the least-square regression line and correlation coefficient ($R$) with the lower and upper bounds for a 95% confidence interval for the coefficient. The $P$-value for testing the hypothesis

that there is no relationship between the $b$ value and SDR (null hypothesis) is indicated by $P$. Since $P$ is smaller than the significance level of 0.05, the corresponding correlation in $R$ was considered to be significant.

We examined if the observed inverse relation between $b$ values and SDRs can happen contingently using the randomized earthquake catalogs. For this examination, a useful measure is the slope of the $b$–SDR relationship. Using the earthquake data to create Fig. 2a, we compared the slope of the observation (slopes of the regression lines for $r = 6$ and 8 km in Fig. 2a) with the slopes based on randomized data sets. A bootstrapping approach to randomization with regard to magnitude was used. In this approach, 200 randomized catalogs were generated, and in each, the slope of the least-square regression line between $b$ values and SDRs was computed. A set of 200 slopes based on randomization was compared with the observed slope for each of the radii $r = 6$ and 8 km. The observed slope was less than most of the simulated slopes for both cases of $r = 6$ and 8 km (Supplementary Figs. 9a, b), and randomized catalogs reproduce the observed slope or steeper at a probability of 0–2%. For comparison, we performed the same analysis for a larger sampling radius $r = 10$ km, and obtained a probability of 12–14% (Supplementary Fig. 9c). The significance level α was set to a typical value of 5% (partitioned to both sides of the distribution as in a two-tailed test, with each tail containing 2.5% of the distribution). We can state that it is very unlikely, when using $r = 6$ and 8 km, that an inverse relation between $b$ values and SDRs in Fig. 2a occurs when the randomized data sets are used.

We adopted an alternative approach by estimating one $b$ value for each sub-fault with one SDR value and plotting $b$ values as a function of SDR. A $b$ value for the closest 200 earthquakes to a sub-fault was computed when the sampling radius was $r \leq 35$ km and only if at least $N_{min} = 50$ events yielded a good fit to the GR law. Results show large scatter plots, although the plots are not inconsistent with the regression lines shown in Fig. 2. We did the same for $N_{min} = 100$ events and obtained similar consistency.

**Data availability**. The JMA earthquake catalog used in this study was obtained from http://www.data.jma.go.jp/svd/eqev/data/bulletin/index.html. Supplementary Table 2 of Yokota et al.[23], showing the list of positions and SDR values, which we used in this study, was downloaded from https://www.nature.com/articles/nature17632#supplementary-information. In creating map images in Fig. 1 and Supplementary Figs. 1a, 2-5, 6a, and 8, we used plate configuration data[40–43], compiled and provided by Meteorological Research Institute. The data are publically available at http://www.mri-jma.go.jp/Dep/sv/2ken/fhirose/en/en.Tools.html. Other information is available from the corresponding author upon reasonable request.

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

## Acknowledgements

We thank Japan Meteorological Agency (JMA) for the JMA earthquake catalog, the Japan Coast Guard for SDR data[23], and Meteorological Research Institute for plate configuration data[40–43] (also see Data availability). Figure 1 and Supplementary Figs. 1a, 2–5, 6a, and 8 were produced by using GMT software[53]. This work was partially supported by JSPS KAKENHI Grant Number JP 17K18958.

## Author contributions

K.Z.N. performed numerical simulations, analyzed data, and prepared the figures. K.Z.N. and A.Y. helped to draft the manuscript and participated in interpretation. K.Z.N. wrote the final manuscript. Both authors read and approved the final manuscript.

## Additional information

**Competing interests:** The authors declare no competing interests.

