## [Peer Review File(PDF 322 kb) · Nature Communications]

Reviewers' comments:

Reviewer #1 (Remarks to the Author):

This ms. contains several important contributions, namely a b-value map of the Nankai subduction interface, and a possible correlation of that with interseismic coupling. However, the presentation is sub-par and I recommend modifications to the authors, described below, that will lead to significant improvement in the paper.

1. the b-value map. The main result is shown in Fig. 1b. I suggest putting Fig. 1a in the supplement section so that a larger version of Fig. 1b can be displayed (the present figs are too small to easily study). There are several issues included in the discussion of this figure that are unrelated to the main topic and require digressions that are both unsatisfying and confusing and that can be deleted. The first is that the low-frequency earthquakes occur below the seismogenic coupled zone – hence the b-values obtained for those depths are unrelated to the main topic, the coupling of the seismogenic interface. They do, however, serve to show the main feature of the b-value map of Fig1b, that b-value decreases with depth. This agrees with the observations of [Spada et al., 2013], and hence the inferred stress dependence of b-value [Scholz, 2015].

However, the orange region off Kii Peninsula is in disagreement with that result. This, however, is a temporal variation associated with some large earthquakes in 2004, as discussed in Fig 2. These earthquakes are in a very unusual location, previously noted (e.g [Yokota et al., 2016] as a site of very long period earthquakes. I suppose these are tsunami earthquakes? In any case this is not a steady-state behavior and should not be included in Fig. 1b. Because there is a whole other issue with regard to temporal changes of b-value around the time of large earthquakes, these observations should be the topic of a separate paper and not included here. I would suggest deleting Fig 2 and all discussion related to it. For this area,(C) perhaps the b-value shown in Fig 2b is a better representative of the long term average and should be the value shown in Fig. 1b.

2. Comparing the coupling map of Yokota et al (fig 2) I see some broad similarities with the b-value map, namely a decreased coupling in the area between the main slip zones of the Nankaido and Tonankai earthquakes. However your plots in Fig. 3 are very confusing. Why do you use log coordinates, the map of Yokota shows only variations between about 3.4 and 5.5 cm/yr, so there is not such a large variation that it needs to be compressed in log coordinates?. For this plot you should not include the b-value data deeper than the couple interface, mentioned above. You also show data in Fig. 3 that is well outside those bounds, including SDRs that greatly exceed the plate velocity! Where did these come from? You need to show a scatter plot of b-value vs SDR in linear coordinates and make a p test to see if it is significant. One possible problem is that the stress dependence of b-value is quite different from the stress dependence of SDR, in which coupling is expected above a certain value of stress and absent below it. To separate these two effects you could try to correlate b-value with SDR over several depth ranges instead of for the entire data set.

Scholz, C. H. (2015), On the stress dependence of the earthquake b value, *Geophysical Research Letters*, 42, doi:10.1002/2014GL062863.

Spada, M., T. Tormann, S. Wiemer, and B. Enescu (2013), Generic dependence of the frequency-size distribution of earthquakes on depth and its relation to the strength profile of the crust, *Geophys. Res. Lett.*, 40, doi:10.1029/2012GL054198.

Yokota, Y., T. Ishikawa, S. Watanabe, T. Tashiro, and A. Asada (2016), Seafloor geodetic constraints on interplate coupling of the Nankai Trough megathrust zone, *Nature*, 534(7607), 374-+, doi:10.1038/nature17632.

Reviewer #2 (Remarks to the Author):

Review of "A b map implying the first eastern rupture of the Nankai Trough earthquakes" by K.Z. Nanjo and A. Yoshida

This manuscript calculates a Gutenberg-Richter b-value map of the Nankai Trough area and discusses spatial patterns. It then correlates the b-values with independent geodetically estimated slip deficit rates, presenting an inverse relationship between the two parameters, which supports and complements observations from previous studies. Combining the slip deficit rates with the relative b-values measured in different parts of the study region, the authors propose the most stressed region to be the eastern part of the 1944/46 earthquakes and argue this to be the most likely nucleation region for the next large event in the region, which would be consistent with the reported rupture history. The importance of b-value imaging has been shown in multiple studies, and this one can become a valuable addition to the field. However, prior to publication some improvements are necessary in the presentation, but also in demonstrating robustness of data, methods and interpretation:

Data & Methods

- The authors stress the importance of M_c assessment in b-value analysis and claim to have carefully checked this. However, there is a striking visual correlation between the gradients in M_c and b (Extended Data Figure 3), i.e. low- M_c /low-b below the coast and increasing M_c and b going offshore towards the trench. To eliminate any doubts whether this is real or biased, a quick additional robustness test can help: two b-value maps for generally increased values of every local M_c by e.g. 0.2 and 0.5 magnitude units, with the expectation that the pattern remains stable (with reduced resolution).
- It remains somewhat unclear from the text, how the sampling for the SDR analysis was performed. How were those b-values calculated, especially with so small radii? According to the 10km b-value map, there are almost no b-values available in the high SDR regions, how can there be enough events with 8 or 6km sampling? Are these b-values really calculated from adding all events sampled around those SDR nodes within the sliding window? But then they are mixtures from different regions with heterogeneous completeness levels – i.e. how stable are those b-value estimates? And how were the window width and step size determined? Would it be possible to estimate one b-value for each SDR node, simply from the closest e.g. 50 or 100 events above M_c , and then provide a scatter plot of SDR versus b? Does this show a similar trend of inverse dependence?
- How were those regression lines selected/computed – they do not span the full window that is cited in the caption?
- It is reasonable to assume that the SDR cannot exceed the plate convergence rate, which according to Yokota et al 2016 and others is 6.5cm/yr. It is thus not clear why here an overshoot limit of 7cm/yr is used. Actually, the 'fit' between b and SDR would become more robust when applying the 6.5cm/yr.
- A very helpful addition would be a figure showing the b-value map and the contour lines of the used SDR data.

Discussion & Interpretation

- The authors observe a low b-value band on the deeper side of the coupled region and call it a 'unique feature' of the Nankai Trough. However, for the interface off Tohoku, low b-values have been shown down to ~100km (Tormann et al, 2015), i.e. also on the down dip side of the focal region. Thus, the two observations seem to be consistent, as are the high b-values observed in both cases near the trench.
- If the occurrence of LFEs is stated to be 'very likely related' (L.83) to patterns of low b-value, it would be nice to include some idea/speculation on what that relation might be.
- The authors identify the interesting feature of higher b-values between the two main rupture patches of the 1944/46 events, i.e. an indication for segmentation? This would be different e.g. from the results for the Pacific Plate in the Tohoku area, and deserves more discussion, e.g.

elaborating on significance with respect to uncertainties and comparing to other studies on asperities and barriers along faults.

- The authors mention an observation from the Parkfield section that showed a significantly different b-value during the period of an aseismic transient event (Schorlemmer et al., 2004). This phenomenon has been studied in more detail by Tormann et al., 2013, who demonstrated a correlation between the time series of b-values and variation in creep rate, interpreting this as changes in loading. Wiemer et al., 2005 studied transients in b-value correlated with changes in subsidence rate in the Tokai region, which also corresponds to changes in locking/slip deficit. Ghosh et al., 2008 equally present the observation of an inverse relation between b-value and locking for the subducting Cocos Plate beneath Costa Rica. Including and comparing with e.g. those or more studies will enrich the discussion in this manuscript.
- Based on the above studies plus the inverse relation between SDR and b observed here, it might even be possible to discuss a slightly higher detection limit in the SDR processing, which would be at ~4cm (lower limit of the observed correlation)? However, this would require a robust verification of the signal, e.g. stability with respect to different sampling/robustness approaches (see comments above).
- The observation that the b-values in the eastern high-SDR regions are lower than in the western counterpart is a neat observation especially together with the rupture history. However, this analysis should be verified based on the M_c /b-value comments for region C (see below), as this region is likely contributing to or even driving the low-b estimates shown in Figure 4.

Figures:

- Unfortunately, most figures in this manuscript are tiny, they need to be significantly increased in size.

It could also prove worthwhile to check what information is really needed on the figures to simplify them, e.g. the volcanoes are not mentioned anywhere in the manuscript and need not complicate the figures.

Furthermore, some captions (e.g. for the maps) are very long, largely repetitive, and consist a lot of information that's relatively obvious with the annotations on the figure. Guiding the reader towards important interpretation or explaining data processing would be better use of the space.

- Figure 1: FMDs for volumes inside the two high slip zones should be added, and for the offshore low-b patch (volume C in Figure 2, see below), those very low b-values of <0.6 are not consistent with the time series in Figure 2
- Figure 2: according to the time series, even in period P2 the b-value fluctuates and is not as low as suggested by the map: are these patches on the maps (P2 and full time period) robust towards increasing M_c ? It could be helpful to replace the stem plot of magnitudes versus time with a scatter plot, as this may show some information on incompleteness, too (also in Extended Data Figure 4).
- Figure 3: The inconsistency between the vertical lines and the limits of the regression lines is not explained and weakens this figure a lot. Adding the correlation coefficient would be valuable. The almost constant result for 40km sampling is irritating, since this level has been chosen as 'optimal' resolution for the spatial mapping with the argument that it does not smooth the spatial variation much (see also comment below)
- Extended Data Figure 1: the differences in colors for the cross sections are not visible – the large events could be added on the cross sections for better orientation.
- Extended Data Figure 2: There seems to be a relatively strong smoothing effect especially in the high slip areas of the 1944/46 events between $r=30$ and $r=40$ – what about alternative sampling approaches, e.g. nearest N events?
- Extended Data Figure 3: The color scale limits for the a-value should match the range of the data.
- Extended Data Figure 4: Is it reasonable that the M_c in frame g should indeed decrease by one magnitude unit following the large mainshock? This drop and the associated low b-values should be verified with showing FMDs from pre-mainshock seismicity, early and later aftershocks.

References:

Ghosh et al. 2008, Interface locking along the subduction megathrust from b-value mapping near Nicoya Peninsula, Costa Rica. GRL, doi:10.1029/2007GL031617

Schorlemmer et al. 2004, Earthquake statistics at Parkfield: 1. Stationarity of b values. JGR, doi:10.1029/2004JB003234

Tormann et al. 2013, Size distribution of Parkfield's microearthquakes reflects changes in surface creep rate. GJI, doi: 10.1093/gji/ggt093

Tormann et al. 2015, Randomness of megathrust earthquakes implied by rapid stress recovery after the Japan earthquake. NGeo, doi: 10.1038/NGEO2343

Wiemer et al., 2005 Correlating seismicity parameters and subsidence in the Tokai region, central Japan. JGR, doi:10.1029/2003JB002732

Yokota et al., 2016 Seafloor geodetic constraints on interplate coupling of the Nankai Trough megathrust zone. Nature, doi:10.1038/nature17632

Response to reviewers' comments

Below please find the comments from the two reviewers (black) as well as our response to them (blue). At the beginning of each comment from reviewer #1, we added a number, such as R1-1, R1-2, etc. Similarly, we added R2-1, R2-2, etc. to reflect responses to comments from reviewer #2. The abbreviations used below are OM: original manuscript and RM: revised manuscript. Major changes that were made have been highlighted in red in the main text. We wish to thank both reviewers for these very valuable comments and suggestions, and trust that the edits made have fully addressed their concerns and requests.

Reviewer #1

This ms. contains several important contributions, namely a b -value map of the Nankai subduction interface, and a possible correlation of that with interseismic coupling. However, the presentation is sub-par and I recommend modifications to the authors, described below, that will lead to significant improvement in the paper.

R1-1: the b -value map. The main result is shown in Fig. 1b. I suggest putting Fig. 1a in the supplement section so that a larger version of Fig. 1b can be displayed (the present figs are too small to easily study). There are several issues included in the discussion of this figure that are unrelated to the main topic and require digressions that are both unsatisfying and confusing and that can be deleted. The first is that the low-frequency earthquakes occur below the seismogenic coupled zone – hence the b -values obtained for those depths are unrelated to the main topic, the coupling of the seismogenic interface. They do, however, serve to show the main feature of the b -value map of Fig1b, that b -value decreases with depth. This agrees with the observations of [Spada et al., 3013], and hence the inferred stress dependence of b -value [Scholz, 2015].

- Based on this useful suggestion, we displayed a larger version of a b map (Fig. 1).
- We believe that the comment “the low-frequency earthquakes occur below the seismogenic coupling zone” may have been a slight misinterpretation by the reviewer. We did not state that low-frequency earthquakes occur just below the seismogenic coupling zone, but we did note that they occur on the deeper side of the coupling zone, where the inter-plate coupling is thought to be very weak or where the subducting and the overriding plates are not seismically

coupled. Subsequently, we suggested that earthquakes in the low- b -value band occur in a drained condition where effective differential stress is likely to be high due to low pore-pressure. This hypothesis is in agreement with the supposition (e.g., Shelly et al., 2006; Ide et al., 2007) that low frequency earthquakes, whose belt overlaps with the zone of the low- b -value band in inner slab earthquakes, occur on the surface of the subducting slab in a condition in which there is plenty of water that is extracted from the slab underneath.

R1-2: However, the orange region off Kii Peninsula is in disagreement with that result. This, however, is a temporal variation associated with some large earthquakes in 2004, as discussed in Fig 2. These earthquakes are in a very unusual location, previously noted (e.g [Yokota et al., 2016] as a site of very long period earthquakes. I suppose these are tsunami earthquakes? In any case this is not a steady-state behavior and should not be included in Fig. 1b. Because there is a whole other issue with regard to temporal changes of b -value around the time of large earthquakes, these observations should be the topic of a separate paper and not included here. I would suggest deleting Fig 2 and all discussion related to it. For this area,(C) perhaps the b -value shown in Fig 2b is a better representative of the long term average and should be the value shown in Fig. 1b.

- We thank the reviewer for this valuable comment. We admit that the low b value off the Kii Peninsula is not a feature of the steady-state, but instead represents a temporal feature that was brought by large earthquakes in 2004. Therefore, Fig. 2 and the discussion related to this topic that appeared in OM were deleted from RM. Our RM intends to clarify the general intrinsic characteristics of the spatial pattern in the b value along the Nankai Trough zone.

R1-3: Comparing the coupling map of Yokota et al (fig 2) I see some broad similarities with the b -value map, namely a decreased coupling in the area between the main slip zones of the Nankaido and Tonankai earthquakes. However your plots in Fig. 3 are very confusing. Why do you use log coordinates, the map of Yokota shows only variations between about 3.4 and 5.5 cm/yr, so there is not such a large variation that it needs to be compressed in log coordinates?. For this plot you should not include the b -value data deeper than the couple interface, mentioned above. You also show data in Fig. 3 that is well outside those bounds, including SDRs that greatly exceed the plate velocity! Where did these come from? You need to show a scatter plot of b -value vs SDR in linear coordinates and make a p test to see if it is significant. One possible problem is that the stress dependence of b -value is quite different from the stress dependence of SDR, in which coupling is expected above a certain value of stress and absent below it. To separate these two effects you could try to correlate

b -value with SDR over several depth ranges instead of for the entire data set.

- We have revised Fig. 3 in OM in response to the reviewer's comment. In the RM, now displayed as Fig. 2, the b value has been plotted against linear values of SDR. In this plot, we used data exclusively from the coupling zone where SDR lies between 0.04 m year^{-1} and $0.058 \text{ m year}^{-1}$, which are values less than the plate convergence rate (Supplementary Fig. 8 indicates areas of high SDRs, whose data was used to obtain the inverse relationship between b -values and SDRs). Even though the correlation between b -values and SDRs is obvious (Fig. 2 and Supplementary Fig. 7), we decided to compute the correlation coefficient and also conducted a p -test. Our results show that the correlation between b and SDR is significant (please see the section "Procedure used to make the graph of b values versus SDRs" in the Methods). To conduct another test, we used randomized catalogs, examined if the observed inverse relationship between b values and SDRs can occur contingently, and found that this is very unlikely to occur (Supplementary Fig. 9).
- In response to another part of the reviewer's comment ("*the stress dependence of b -value is quite different from the stress dependence of SDR, in which coupling is expected above a certain value of stress and absent below it*"), we decided to discuss the correlation between b and SDR for a range of higher SDRs ($\geq 0.04 \text{ m year}^{-1}$), even though the detection limit of SDR is 0.03 m year^{-1} (Fig. 2). As indicated in our response R1-1, we believe that the plate interface coupling in the deeper regions is very weak. Moreover, as described in the "Results" section of RM, differential stress is thought to be low near the trough axis. In addition, if we consider regions of SDRs to be $\geq 0.04 \text{ m year}^{-1}$, excluding regions north of 34.5°N , we can observe that the variation in SDRs is mainly located in, and restricted to, an intermediate depth range (10-30 km). Thus, because this depth range is considerably narrow, we think that it is meaningless to correlate the b -value with SDR over several depth ranges.

Reviewer #2

This manuscript calculates a Gutenberg-Richter b -value map of the Nankai Trough area and discusses spatial patterns. It then correlates the b -values with independent geodetically estimated slip deficit rates, presenting an inverse relationship between the two parameters, which supports and complements observations from previous studies. Combining the slip deficit rates with the relative b -values measured in different parts of the study region, the authors propose the most stressed region to be the eastern part of the 1944/46 earthquakes and argue this to be the most likely nucleation region for the next large event in the region, which would be consistent with the reported rupture

history. The importance of b-value imaging has been shown in multiple studies, and this one can become a valuable addition to the field. However, prior to publication some improvements are necessary in the presentation, but also in demonstrating robustness of data, methods and interpretation:

Data & Methods

R2-1: • The authors stress the importance of M_c assessment in b-value analysis and claim to have carefully checked this. However, there is a striking visual correlation between the gradients in M_c and b (Extended Data Figure 3), i.e. low- M_c /low-b below the coast and increasing M_c and b going offshore towards the trench. To eliminate any doubts whether this is real or biased, a quick additional robustness test can help: two b-value maps for generally increased values of every local M_c by e.g. 0.2 and 0.5 magnitude units, with the expectation that the pattern remains stable (with reduced resolution).

- In response to these comments, we confirmed that the main features of the b value map were retained when M_c was increased by 0.2 and by 0.5. (Please see the section “Mapping procedure” of Methods as well as Supplementary Fig. 5).

R2-2: • It remains somewhat unclear from the text, how the sampling for the SDR analysis was performed. How were those b-values calculated, especially with so small radii? According to the 10km b-value map, there are almost no b-values available in the high SDR regions, how can there be enough events with 8 or 6km sampling? Are these b-values really calculated from adding all events sampled around those SDR nodes within the sliding window? But then they are mixtures from different regions with heterogeneous completeness levels – i.e. how stable are those b-value estimates? And how were the window width and step size determined? Would it be possible to estimate one b-value for each SDR node, simply from the closest e.g. 50 or 100 events above M_c , and then provide a scatter plot of SDR versus b? Does this show a similar trend of inverse dependence?

- We calculated b values by sampling and adding all events within an assigned radius, such as 6 or 8 km, from all nodes, around those SDR nodes, within the sliding window (Fig. 2 and Supplementary Fig. 7).
- We appreciate the reviewer’s comment, and understood that the b values are mixtures from different regions with heterogeneous levels of completeness. To remove the effect of these mixtures from our analysis, we pre-cut the earthquake catalog at $M_{2.2}$ (to apply the

completeness catalog for the entire Nankai Trough in Supplementary Fig. 3c, one needs to use earthquakes with $M_{2.2}$ or larger). We then computed b values (Fig. 2 and Supplementary Fig. 7).

- For window size, we considered two window widths: 0.014 and 0.01 m year^{-1} (Please see Fig. 2, section “Procedure used to make the graph of b values versus $SDRs$ ” of the Methods, as well as Supplementary Fig. 7). We found that the general pattern in both cases remained similar, indicating that the effect of window width on the results is negligible. For step size, we decided to use 0.001 m year^{-1} because this value is the significant figure of the SDR dataset that can be found in the Supplementary Information of the Yokota et al. (2015) study. We did not consider smaller values of step size than the value used.
- In response to the request in the reviewer’s comment, we estimated one b -value for each SDR node, simply from the closest (e.g. 50 or 100) events above M_c , and then created a plot of SDR versus b . As reviewer #2 pointed out, this plot is scattered but is not inconsistent with the regression lines shown in Fig. 2, as described in the section “Procedure used to make the graph of b values versus $SDRs$ ” of the Methods.

R2-3: • How were those regression lines selected/computed – they do not span the full window that is cited in the caption?

- The least-square regression lines were computed for data points 0.04-0.058 m year^{-1} (Please see R1-3). The equations of the least-square regression lines with errors are provided in Fig. 2. The correlation coefficient R with the lower and upper bounds in brackets are also shown in Fig. 2. Furthermore, the p -value for testing the hypothesis that there is no relationship between the b -value and SDR (our null hypothesis) is indicated by P in Fig. 2. Please also see the section “Procedure used to make the graph of b values versus $SDRs$ ” of the Methods and Supplementary Fig. 7.

R2-4: • It is reasonable to assume that the SDR cannot exceed the plate convergence rate, which according to Yokota et al 2016 and others is 6.5cm/yr. It is thus not clear why here an overshoot limit of 7cm/yr is used. Actually, the ‘fit’ between b and SDR would become more robust when applying the 6.5cm/yr.

- Thank you for this pertinent comment and/or query. We assumed that SDR cannot exceed the plate convergence rate of 0.065 m year^{-1} (Please see Fig. 2).

R2-5: • A very helpful addition would be a figure showing the b -value map and the contour lines of the used SDR data.

- This was an excellent suggestion. We added a figure (Supplementary Fig. 8) that shows the b -value map and the contour lines of the SDR data used.

Discussion & Interpretation

R2-6: • The authors observe a low b -value band on the deeper side of the coupled region and call it a ‘unique feature’ of the Nankai Trough. However, for the interface off Tohoku, low b -values have been shown down to ~ 100 km (Tormann et al, 2015), i.e. also on the down dip side of the focal region. Thus, the two observations seem to be consistent, as are the high b -values observed in both cases near the trench.

- We suggested that earthquakes in the low- b -value band occur in a drained condition where effective differential stress should be high due to low pore-pressure. This hypothesis is also in agreement with the supposition (e.g., Shelly et al., 2006; Ide et al., 2007) that low frequency earthquakes, whose belt overlaps with the zone of the low- b -value band in inner slab earthquakes, occur on the surface of the subducting slab in a condition in which there is plenty of water that is extracted from the slab underneath. We considered that “the low- b -value belt” is unique, implying that it overlaps with the zone of low-frequency earthquakes, unlike the interface off-Tohoku, where low-frequency earthquakes are not observed on the downdip side of the coupling region.
- According the reviewer’s last comment, we understand that high b values near the trench show a common feature of both the Nankai Trough and the Tohoku-oki cases (Please see “Results”).

R2-7: • If the occurrence of LFEs is stated to be ‘very likely related’ (L.83) to patterns of low b -value, it would be nice to include some idea/speculation on what that relation might be.

- We considered that slab earthquakes in the low- b -value band occur in a drained condition where effective differential stress is likely to be high due to low pore-pressure. This notion is in agreement with the supposition that LFEs, whose belt overlaps with the low- b -value band, occur on the surface of the subducting slab in the condition of plentiful waters that are extracted from the slab underneath. Please see also read our response in R2-6.

R2-8: • The authors identify the interesting feature of higher b -values between the two main rupture patches of the 1944/46 events, i.e. an indication for segmentation? This would be different e.g. from

the results for the Pacific Plate in the Tohoku area, and deserves more discussion, e.g. elaborating on significance with respect to uncertainties and comparing to other studies on asperities and barriers along faults.

- Based on this suggestion, we added a figure (Supplementary Fig. 6) that shows changes in the b value along the line that passes through focal regions of the 1944 Tonankai and the 1946 Nankai earthquakes. That figure certifies that a high- b -value zone exists around 135°E which separates low- b -value areas corresponding to the main patches of the two earthquakes.
- To address the above comment, we added the following text to the “Results”: *Existence of the two notable low- b -value patches as well as the high- b -value area between them is unique in the Nankai Trough subduction zone, suggesting that the plate interface is segmented in such a way that two mega-thrust earthquakes might occur separately, thereby breaking each patch. This spatial pattern of the b value is in contrast with that in the Tohoku-oki subduction zone (Tormann et al., 2015), where the Pacific Plate subducts beneath the North-American plate. We also compared these findings to other studies on asperities and barriers along faults, and expanded the “Discussion” as follows: Wiemer et al. (2005) demonstrated that transients in subsidence rate in the Tokai region, specifically the eastern Nankai Trough zone, were correlated with a change in the occurrence rate and size distribution of earthquakes, i.e., the b value. They surmised that an increase in the locking stress, namely a “slow stick event”, was consistent with all observations. Ghosh et al. (2008) observed an inverse relation between the b value and locking for the subducting Coos Plate beneath Costa Rica. An observation from the Parkfield segment of the San Andreas fault showed significantly different b values during a period of aseismic transient events, corresponding to unlocking a part of that segment (Schorlemmer et al., 2005). This phenomenon was studied in more detail by Tormann et al. (2013) who demonstrated a correlation between the time-series of b values and variation in creep rate. When all of these observations are taken into account, it is reasonable to consider that the low b value in areas of large SDR indicates that differential stress is large on asperities.*

R2-9: • The authors mention an observation from the Parkfield section that showed a significantly different b -value during the period of an aseismic transient event (Schorlemmer et al., 2004). This phenomenon has been studied in more detail by Tormann et al., 2013, who demonstrated a correlation between the time series of b -values and variation in creep rate, interpreting this as changes in loading. Wiemer et al., 2005 studied transients in b -value correlated with changes in

subsidence rate in the Tokai region, which also corresponds to changes in locking/slip deficit. Ghosh et al., 2008 equally present the observation of an inverse relation between b -value and locking for the subducting Cocos Plate beneath Costa Rica. Including and comparing with e.g. those or more studies will enrich the discussion in this manuscript.

- We deleted Fig. 2 and related discussion on this topic that appeared in OM (Please also see R1-2). The RM intends to clarify the general intrinsic characteristics of the spatial pattern of the b value along the Nankai Trough. However, since we believe that reviewer #2's comment is important, we included a comment in the "Discussion" that referred to Tormann et al. (2013), Wiemer et al. (2005) and Ghosh et al. (2008), and also added these to the "References" (Please also see R2-8).

R2-10: • Based on the above studies plus the inverse relation between SDR and b observed here, it might even be possible to discuss a slightly higher detection limit in the SDR processing, which would be at $\sim 4\text{cm}$ (lower limit of the observed correlation)? However, this would require a robust verification of the signal, e.g. stability with respect to different sampling/robustness approaches (see comments above).

- This was an excellent suggestion. Based on this suggestion, we decided to employ a higher value (0.04 m year^{-1}) than the detection limit of SDR (0.03 m year^{-1}) as the lower limit of the range of $SDRs$ (Please see Supplementary Fig. 8 for areas of $SDR \geq 0.04 \text{ m year}^{-1}$). We also conducted robust verification of the signal as can be seen in Supplementary Figs. 2-7 and in the section "Procedure used to make the graph of b values versus $SDRs$ " of the Methods). We hope that this verification satisfies reviewer #2.

R2-11: • The observation that the b -values in the eastern high- SDR regions are lower than in the western counterpart is a neat observation especially together with the rupture history. However, this analysis should be verified based on the M_c/b -value comments for region C (see below), as this region is likely contributing to or even driving the low- b estimates shown in Figure 4.

- We deleted the discussion regarding the temporal change in b because we agreed with the comment (R1-2) made by reviewer #1. A temporal change in region C that appeared in OM was not included in the RM. We modified Fig. 4 in OM, and in the new version shown in Fig. 3 in RM, the effect of temporal change in b was not included.

Figures:

R2-12: • Unfortunately, most figures in this manuscript are tiny, they need to be significantly increased in size. It could also prove worthwhile to check what information is really needed on the figures to simplify them, e.g. the volcanoes are not mentioned anywhere in the manuscript and need not complicate the figures. Furthermore, some captions (e.g. for the maps) are very long, largely repetitive, and consist a lot of information that's relatively obvious with the annotations on the figure. Guiding the reader towards important interpretation or explaining data processing would be better use of the space.

- In response to this comment, we modified all figures and captions. Figures are no longer tiny in RM. Furthermore, we checked whether information in the figures was absolutely necessary, and shortened the captions where necessary. We hope that the overall modifications satisfy reviewer #2.

R2-13: • Figure 1: FMDs for volumes inside the two high slip zones should be added, and for the offshore low- b patch (volume C in Figure 2, see below), those very low b -values of <0.6 are not consistent with the time series in Figure 2

- To address this comment, we added the FMDs for volumes inside the two high slip zones in Fig. 1 of RM.
- In RM, we deleted Fig. 2 and the discussion related to this topic that appeared in OM (Please also see R1-2). RM intends to clarify the general intrinsic characteristics of the spatial pattern of the b value along the Nankai Trough. The effect of temporal change in b in region C that appeared in OM was not included in the RM (Please also see R2-11).

R2-14: • Figure 2: according to the time series, even in period P2 the b -value fluctuates and is not as low as suggested by the map: are these patches on the maps (P2 and full time period) robust towards increasing M_c ? It could be helpful to replace the stem plot of magnitudes versus time with a scatter plot, as this may show some information on incompleteness, too (also in Extended Data Figure 4).

- As discussed in R1-2, R2-11, and R2-13, the temporal change in b in the OM was not included in the RM. We removed the time series in b values and the magnitude-time plot from the RM.

R2-15: • Figure 3: The inconsistency between the vertical lines and the limits of the regression lines is not explained and weakens this figure a lot. Adding the correlation coefficient would be valuable. The almost constant result for 40km sampling is irritating, since this level has been chosen as

‘optimal’ resolution for the spatial mapping with the argument that it does not smooth the spatial variation much (see also comment below)

- To address the request in this comment, we modified Fig. 2 of the RM. For details of the procedure used to make the graph of b values versus $SDRs$, please see the Methods. Related responses are also provided in R1-3 and R2-10.
- We also removed the almost constant result for 40 km sampling in Fig. 3 of the OM, from Fig. 2 of the RM.

R2-16: • Extended Data Figure 1: the differences in colors for the cross sections are not visible – the large events could be added on the cross sections for better orientation.

- To address these requests, we fortified the differences in colors for the cross sections by making them more visible, as shown in Supplementary Fig. 1. We also added the large events to Supplementary Fig. 1.

R2-17: • Extended Data Figure 2: There seems to be a relatively strong smoothing effect especially in the high slip areas of the 1944/46 events between $r=30$ and $r=40$ – what about alternative sampling approaches, e.g. nearest N events?

- To address this query, we reconsidered sampling radius optimization (Please also see “Mapping procedure” of the Methods) and assumed $r = 35$ km as the optimal radius. We checked an alternative mapping approach, by sampling nearest N events to each node (Supplementary Fig. 4), which indicated that the general pattern remains stable with regards to sampling methods (for example, for both mapping approaches, we can see the contrast of low b values in the high slip areas versus high b values in between).

R2-18: • Extended Data Figure 3: The color scale limits for the a -value should match the range of the data.

- We modified Supplementary Fig. 3b so that the color scales match the ranges of the data.

R2-19: • Extended Data Figure 4: Is it reasonable that the M_c in frame g should indeed decrease by one magnitude unit following the large mainshock? This drop and the associated low b -values should be verified with showing FMDs from pre-mainshock seismicity, early and later aftershocks.

- Similar to R1-2, R2-11, R2-13, and R2-14, the temporal change in b and M_c discussed in the OM was not included in the RM. We decided not to discuss the related time-series in b and M_c .

in the RM. We decided to examine the spatial features of the b value based on the data after 2006. One reason is that the *SDR* analysis was performed by using data in that period (Please see the section “Data processing of *SDRs*” of the Methods). This choice was also preferred for the purpose of investigating intrinsic features of the b value, because no large earthquake that affected these features occurred in that period.

References

1. Ide, S., Shelly, D. R. & Beroza, G. C. Mechanism of deep low frequency earthquakes: Further evidence that deep non-volcanic tremor is generated by shear slip on the plate interface. *Geophys. Res. Lett.* **34**, L03308 (2007).
2. Ghosh, A., Newman, A. V., Thomas, A. M. & Farmer, G. T. Interface locking along the subduction megathrust from b -value mapping near Nicoya Peninsula, Costa Rica. *Geophys. Res. Lett.* **35**, L01301 (2008).
3. Schorlemmer, D., Wiemer, S. & Wyss, M. Earthquake statistics at Parkfield: 1. Stationarity of b values. *J. Geophys. Res.* **109**, B12307 (2004).
4. Shelly D. R., Beroza G. C., Ide S. & Nakamura, S. Low-frequency earthquakes in Shikoku, Japan, and their relationship to episodic tremor and slip. *Nature* **442**, 188-191 (2006).
5. Tormann T., Wiemer S., Metzger S., Michael A., Hardebeck J. L. Size distribution of Parkfield’s microearthquakes reflects changes in surface creep rate. *Geophys. J. Int.* **193**, 1474-1478 (2013).
6. Tormann, T., Enescu, B., Woessner J. & Wiemer, S. Randomness of megathrust earthquakes implied by rapid stress recovery after the Japan earthquake. *Nature Geosci.* **8**, 152-158 (2015).
7. Wiemer S., Yoshida A., Hosono K., Noguchi, S. & Takayama, H. Correlating seismicity parameters and subsidence in the Tokai region, central Japan. *J. Geophys. Res.* **111**, B10303 (2005).
8. Yokota, Y., Ishikawa, T., Watanabe, S., Tashiro, T. & Asada, A. Seafloor geodetic constraints on interplate coupling of the Nankai Trough megathrust zone. *Nature* **534**, 374–377 (2016).

REVIEWERS' COMMENTS:

Reviewer #2 (Remarks to the Author):

Review of the revised manuscript "A b map implying the first eastern rupture of the Nankai Trough earthquakes" by K.Z. Nanjo and A. Yoshida

The revised manuscript, supplement and point-by-point responses address all raised comments thoroughly and conclusively. The presentation of the study improved greatly: the manuscript is now very focused and illustrated by clear and convincing figures. It provides comprehensible information on methods and the additionally performed sensitivity checks confirm the robustness of the presented findings. This paper is a valuable contribution to the field of b-value analyses.

NCOMMS-17-21388A

Response to reviewer's comment

Below please find the comment from the reviewer (black) as well as our response to it (blue). We wish to thank the reviewer for his/her very valuable comment.

Reviewer #2

The revised manuscript, supplement and point-by-point responses address all raised comments thoroughly and conclusively. The presentation of the study improved greatly: the manuscript is now very focused and illustrated by clear and convincing figures. It provides comprehensible information on methods and the additionally performed sensitivity checks confirm the robustness of the presented findings. This paper is a valuable contribution to the field of b-value analyses.

- We thank the reviewer for this valuable comment. We added an additional comment to the “Results” (Lines 72-73). Furthermore, we completed a thorough final revision of the entire text.